# YOLO Series for Human Hand Action Detection and Classification from Egocentric Videos

**DOI:** 10.3390/s23063255

**Published:** 2023-03-20

**Authors:** Hung-Cuong Nguyen, Thi-Hao Nguyen, Rafał Scherer, Van-Hung Le

**Affiliations:** 1Faculty of Engineering Technology, Hung Vuong University, Viet Tri City 35100, Vietnam; 2Department of Intelligent Computer Systems, Czestochowa University of Technology, 42-218 Czestochowa, Poland; 3Faculty of Basic Science, Tan Trao University, Tuyen Quang City 22000, Vietnam

**Keywords:** hand detection, hand classification, YOLO-family networks, convolutional neural networks (CNNs), egocentric vision

## Abstract

Hand detection and classification is a very important pre-processing step in building applications based on three-dimensional (3D) hand pose estimation and hand activity recognition. To automatically limit the hand data area on egocentric vision (EV) datasets, especially to see the development and performance of the “You Only Live Once” (YOLO) network over the past seven years, we propose a study comparing the efficiency of hand detection and classification based on the YOLO-family networks. This study is based on the following problems: (1) systematizing all architectures, advantages, and disadvantages of YOLO-family networks from version (v)1 to v7; (2) preparing ground-truth data for pre-trained models and evaluation models of hand detection and classification on EV datasets (FPHAB, HOI4D, RehabHand); (3) fine-tuning the hand detection and classification model based on the YOLO-family networks, hand detection, and classification evaluation on the EV datasets. Hand detection and classification results on the YOLOv7 network and its variations were the best across all three datasets. The results of the YOLOv7-w6 network are as follows: FPHAB is *P* = 97% with *Thesh_IOU_* = 0.5; HOI4D is *P* = 95% with *Thesh_IOU_* = 0.5; RehabHand is larger than 95% with *Thesh_IOU_* = 0.5; the processing speed of YOLOv7-w6 is 60 fps with a resolution of 1280 × 1280 pixels and that of YOLOv7 is 133 fps with a resolution of 640 × 640 pixels.

## 1. Introduction

Building an application to support the rehabilitation of the hand after surgery is an issue of interest in artificial intelligence, machine learning, deep learning, and computer vision. The quantification of the patient’s hand function after surgery was previously only based on the subjectivity of the doctors. To have objective and accurate assessments and exercise orientation for patients, it is necessary to support an assessment system. Through the research process, we propose a model to build a help system, as illustrated in Figure 1. Figure 1 includes three steps: Input—image sequence from EV; hand tracking detection; estimate 2D, 3D hand pose; hand activities recognition; Output—quantification sums up the results to show the action ability of the hand. An EV dataset refers to a dataset that is collected from the perspective of a single individual, usually with the use of wearable cameras or other devices that record the individual’s view of their surroundings. These datasets typically include video and audio, and they may be used in a variety of applications, such as computer vision, human–computer interaction, and virtual reality. Figure 1 presents hand detection as an important pre-processing step in the application construction process; the detected hand data area is very decisive to the estimation space of the 2D hand pose and 3D hand pose. The problem of hand detection is not a new study; however, the problem persists when it comes to detection in the EV datasets. Since the fingers are obscured by the direction of view or other objects, the visible data area is only the back of the hand. Often, studies using deep learning models for 3D hand pose estimation and hand activity recognition apply third-person viewpoint datasets such as the NYU [1], ICVL [2], and MSRA [3] datasets. These datasets usually have segmented hand data with the environment and hand data not obfuscated and lost data of the fingers, as illustrated in Figure 2.

During the research, we performed a study using Google Mediapipe (GM) [5,6] for hand detection and classification [7] on the HOI4D [8] dataset. The results show that the pre-trained models of models have low results in hand detection and classification (Tables 1 and 2 [7]). Figure 3 shows some cases where the hand is not detected when using the GM on the FPHAB and HOI4D datasets.

Recently, the YOLOv7 model was proposed by Wang et al. [9]. YOLOv7-E6 [9] is more accurate and faster than SWINL Cascade-Mask R-CNN [10] by 2% and 509%, respectively. YOLO v7 is more accurate and faster than other versions of YOLO such as YOLOR [11], YOLOX [12], Scaled-YOLOv4 [13], YOLOv5 [14], DETR [15], and Deformable DETR [16].

In this paper, we are interested in the “hand detection and classification” pre-processing step. We propose using YOLO-family networks with their advantages of accuracy and processing speed for fine-tuning the pre-trained model to detect and classify hand action on many different EV datasets (FPHAB [4], HOI4D [8], RehabHand) with many contexts and different hand movements. FPHAB [4] and HOI4D [8] datasets are two datasets collected from EV and published to evaluate 2D and 3D hand pose estimation models. The RehabHand dataset is also collected from EV mounted on patients who practice grasping rehabilitation at Hanoi Medical University Hospital, Vietnam.

The main contributions of the paper are as follows:A framework for building an image-based rehabilitation evaluation system of EV is proposed.We systematize the architectures of the YOLO-family networks for object detection.We fine-tune hand action detection and classification of the model based on the YOLO-family networks on the first-person viewpoint/EV datasets (FPHAB [4], HOI4D [8], RehabHand [17]).We manually mark the hand data area in the datasets for the evaluation of the hand detection results on the FPHAB [4], HOI4D [8], and RehabHand [17] datasets.Experiments on hand action detection and classification are presented in detail, and the results of hand action detection and classification are evaluated and compared with YOLO-family networks on the FPHAB [4], HOI4D [8], and RehabHand [17] datasets.

The content of this paper is organized as follows: Section 1 introduces the applications and difficulties of hand action detection and classification on the EV datasets. Section 2 discusses related research in this field. Section 3 presents the process of applying YOLO-family networks to fine-tune the hand action detection and classification models. Section 4 compares the models quantitatively and shows qualitative experiments. Section 5 concludes the contributions and presents the future works.

## 2. Related Works

The problem of hand detection and classification is not a new research direction. However, the results of hand detection are very important in the process of building applications for human–machine interaction or building support systems. However, when detecting the hand on datasets obtained from EV, there are still some challenges caused by the external conditions, such as fingers being completely obscured due to the viewpoint of the camera and obscured by objects when grasping the object, where the image obtained only has data of the hand palm. There are also now several EV datasets to evaluate computer vision studies. Recently, Marcos et al. [18] published a helpful survey research on activity recognition on EV datasets.

Ren et al. [19] proposed a dataset called Intel EV with 10 video sequences. The total amount of video data is about 120 min, with 100,000 frames; about 70,000 frames contain objects, with about 1600 per object and 42 different hand actions. Fathi et al. [20] published a database under the name GTEA Gaze with more than 30 different types of food and objects. GTEA Gaze includes 94 types of actions and 33 classes of objects. There are also some typical databases collected from EV such as H_2_O [21], Meccano [22], etc.

Particularly, Bandini et al. [23] analyzed problems of computer vision based on an EV dataset. The authors focused on three main research directions: localization (hand detection, hand segmentation, hand pose estimation), interpretation (hand gesture recognition, grasping object, hand action recognition, hand activity recognition), and application (hand-based human–computer interaction, healthcare application). The three research directions explored in Bandini et al.’s [23] paper constitute a unified process with the output applications based on hand data obtained from EV.

Today, with the development of computer hardware and the advent of deep learning, researchers have become equipped with novel tools, the most prominent of which are various convolutional networks (CNN). There have been many published CNN-based researches on hand detection such as YOLOv1 [24], YOLOv2 [25], YOLOv3 [26], YOLOv4 [27], YOLOv5 [14,28], YOLOv7 [9], Mask R-CNN [29,30], SSD [31], MobileNetv3 [32], etc. Some of the most prominent results are shown in Figure 1 of Wang et al.’s work [9], where YOLOv7 achieved the best results in terms of accuracy and speed.

More specifically, a study by Gallo et al. [33] used YOLOv7 to evaluate the detection of weeds near plants using images collected from UAVs. The results of the weeds are a mAP@0.5 score of 56.6%, recall of 62.1%, and precision of 61.3%. Huang et al. [34] used YOLOv3 to detect and determine the patient’s venous infusion based on flow waveforms. The results were compared with RCNN and Fast-RCNN. Detection results showed a precision of 97.68% and recall of 96.88%. Liu et al. [35] used the YOLOv3 model with four-scale detection layers (FDL) to detect combined B-scan and C-scan GPR images. The proposed method can detect both large particles and small cracks. Recently, Lugaresi et al. [5] and Zhang et al. [6] proposed and evaluated a Mediapipe framework that can perform hand detection on both a CPU and GPU with 95.7% detection accuracy with all types of hand palms in real life.

## 3. Hand Action Detection and Classification Based on YOLO-Family Networks

### 3.1. YOLO-Family Networks for Object Detection

Object detection is an important problem in computer vision. YOLO is a convolutional neural network rated with average accuracy; however, the computation speed is very fast and the computation can be performed on a CPU [36]. As studied by Huang et al. [36], when evaluated on the Pascal VOC 2012 dataset, the accuracy results of R-FCN [37], Faster R-CNN [38], SSD [39], and YOLOv3 [26] are 80.5%, 70.4%, 78.5%, and 78.6%, respectively. The processing time results of R-FCN [37], Faster R-CNN [38], SSD [39], and YOLOv3 [26] are 6 fps, 17 fps, 59 fps, and 91 fps, respectively. Before YOLO was born, there were some CNNs such as R-CNN, Fast R-CNN, and Faster R-CNN using a two-stage detector method that obtained very impressive accuracy results but high computation time. To solve the computation time problem, YOLO uses one-stage detectors for object detection.

YOLO version 1 (YOLOv1) [24] uses 24 convolutional layers: 1 × 1 reduction layers (used to reduce image size) followed by 3 × 3 convolutional layers, and max pooling layers. The architecture ends with two fully connected layers. The result is a three-dimensional matrix of size 7 × 7 × 30, as illustrated in Figure 4.

YOLO divides the image into S×S cells, with each cell being a matrix A. If the center of the object is in the cell (i,j), the corresponding output will be in A[i,j]. The prediction process is performed in two steps as follows: the convolutional network performs the feature extraction of the images; extra layers (fully connected layers) analyze and detect the object, then return the output as a matrix A of the following size:(1)size(A)=S×S×(5×B+C)
where *B* is the number of bounding boxes; each bounding box has five components: (x,y,w,h,CS). Confidence Score CS is the probability that the cell contains an object. Finally, the *C* elements are the representation of the probability distribution of the class object. This *C* element is a probability distribution $p_{i}$ and satisfies
(2)$$\sum_{i=0}^{c}p_{i}=1$$

Loss function: YOLO uses the Sum-Squared Error (SSE) function. The values x,y,w,h,*C* are the values of the ground truth box, and the values $\tilde{x},\tilde{y},\tilde{w},\tilde{h},\tilde{C}$ are the predicted bounding box.
(3)$$SSE=E_{1}+E_{2}+E_{3}+E_{4}+E_{5}$$
where
(4)$$E_{1}=\lambda_{coord}\sum_{i=0}^{S^{2}}\sum_{j=0}^{B}{\mathrm{LF}}_{ij}^{object}\left[\;{\left(x_{i}-{\tilde{x}}_{i}\right)}^{2}+{\left(y_{i}-{\tilde{y}}_{i}\right)}^{2}\right]\;$$
(5)$$E_{2}=\lambda_{coord}\sum_{i=0}^{S^{2}}\sum_{j=0}^{B}{\mathrm{LF}}_{ij}^{object}[\;{\left(\sqrt{w_{i}}-\sqrt{{\tilde{x}}_{i}}\right)}^{2}+\left(\sqrt{h_{i}}-{\sqrt{{\tilde{h}}_{i})}}^{2}\right]\;$$
(6)$$E_{3}=\lambda_{coord}\sum_{i=0}^{S^{2}}\sum_{j=0}^{B}{\mathrm{LF}}_{ij}^{object}{\left(C_{i}-{\tilde{C}}_{i}\right)}^{2}$$
(7)$$E_{4}=\lambda_{no\_object}\sum_{i=0}^{S^{2}}\sum_{j=0}^{B}{\mathrm{LF}}_{ij}^{no\_object}{\left(C_{i}-{\tilde{C}}_{i}\right)}^{2}$$
(8)$$E_{5}=\sum_{j=0}^{B}{\mathrm{LF}}_{i}^{object}\sum_{c\in classes}{\left(p_{i}\left(c\right)-\tilde{p_{i}}\left(c\right)\right)}^{2}$$
where $E_{1}$ is xy_loss when the object exists at $box_{j}$ in $cell_{i}$;

$E_{2}$ is wh_loss when the object exists at $box_{j}$ in $cell_{i}$;

$E_{3}$ is confidence_loss when the object exists at $box_{j}$ in $cell_{i}$;

$E_{4}$ is confidence_loss when objects do not exist in the boxes;

$E_{5}$ is class_probability_loss in the cell where the object exists.

Further, ${\mathrm{LF}}_{ij}^{object}=1$ if in the $i^{th}$ cell, there is a $j^{th}$ box containing an object;

${\mathrm{LF}}_{ij}^{no\_object}$ is the opposite of ${\mathrm{LF}}_{ij}^{object}$;

${\mathrm{LF}}_{ij}^{object}=1$ if the $i^{th}$ cell contains an object (otherwise, it is 0);

$\lambda_{coord}$, $\lambda_{no\_object}$ is the component weight.

However, even a good model still has a case: predicting multiple bounding boxes for the same object. To solve this problem, YOLO filters out redundant bounding boxes (duplicate and same class) by non-maximum suppression with two steps as follows:-Boxes with confidence_score are ranked from high to low [box_0, box_1, ⋯, box_n].-Traverse from the top of the list, for each box_i, removing box_j that have IOU(box_i,box_j)≥ threshold, where j>i. The threshold is a pre-selected threshold value. IOU is the formula for calculating the overlap–interference between two bounding boxes, as computed in Equation (Equation 10).

YOLOv2 [25] was born to improve on the weaknesses of YOLOv1 [24]. YOLOv2 makes the following improvements:Batch Normalization (BN): adding BN to all convolutional layers. This allows weights that would never have been learned without BN to be learned again, and reduces the dependence on the initialization of parameter values.High-Resolution Classifier: training the classifier with 224 × 224 and training with 448 × 448 at least 10 epochs for object detection.Anchor Box: they are pre-generated bounding boxes (not model-predicted bounding boxes). With a grid, it creates some *K* anchor boxes with different sizes. These anchor boxes will predict whether it contains an object or not, based on the results of the calculation of the IOU between it and the ground truth (if the IOU>50%, the anchor box is considered to contain the object). Figure 5 shows the process of using anchor boxes for object prediction in an image. YOLOv2 divides the image into 13 × 13 grid cells; so, the ability to find small objects is higher than that of YOLOv1, which is 7 × 7. YOLOv2 is trained on images that vary in size from 320 × 320 up to 640 × 640. This enables the model to learn more features of the object and have higher accuracy. YOLOv2 uses the Darknet19 with 19 convolution layers along with 5 max-pooling layers (it does not use fully connected layers for prediction but anchor boxes instead). Without using fully connected classes and using anchor boxes instead, the final result of the model will be 13 × 13 × 125. For each tensor of size 1 × 1 × 125, it is calculated as follows: k× (5 + 20), where k=5 and 20 is the number of pre-trained object classes. Darknet19 is very fast in object recognition; thus, it makes a lot of sense for real-time processing. The architecture is presented in Figure 6.

YOLOv3 [26] was born to improve on the weaknesses of YOLOv1 [24] and YOLOv2 [25]. YOLOv3 uses Darknet53 as the backbone (with 53 convolutional layers), as illustrated in Figure 7. YOLOv3 performs recognition three times on an image with different sizes. YOLOv3 has its output changed to S×S×255, with *S* being the values 13, 26, and 52, respectively. With each grid box, there are nine different anchor boxes with sizes: grid cell 13 × 13: (116 × 90), (156 × 198), (373 × 326); grid cell 26 × 26: (30 × 61), (62 × 45), (59 × 119); grid cell 52 × 52: (10 × 13), (16 × 30), (33 × 23). The training process combines with the k-means clustering algorithm and uses the ground truth to calculate the error between the ground truth and the anchor box by adjusting values (x,y,w,h), thereby learning the features of the object.

YOLOv4 [27] was developed to improve the accuracy and processing time of YOLOv3 [26]. YOLOv4 applies the idea of CSPBlock, replaces the usual Residual Block of YOLOv3 to CSPResBlock, and also changes the activation function from LeakyReLU to Mish, creating CSPDarkNet53. The structure of YOLOv4 is divided into four parts:Backbone: The backbone can be selected from one of the following three backbones: CSPResNext50, CSPDarknet53, and EfficientNet-B3. CSPDarknet53 is built on a combination of CSP (Cross-Stage-Partial connections) and Darknet53.The CSP is derived from the DenseNet architecture that takes the previous input and concatenates it with the current input before moving into the Dense layer. The role of CSP is to remove computational bottlenecks in DenseNet and improve learning by porting an unmodified version of the feature map. DenseNet (Dense-connected convolutional network) is one of the latest networks for visual object recognition. Densenet has a structure of dense blocks and transition layers. With traditional CNN, if we have *L* layers, there will be *L* connections; however, in DenseNet, there will be L(L+1)/2 connections (i.e., the front layer will be connected with all the layers behind it). Yolov4 uses CSPDarknet53 as the backbone.The main idea of CSPBlock of CSPDarknet53 is applied to Residual Block, as presented in Figure 8 [13]. Instead of having only one path from beginning to end, CSPBlock is divided into two paths. By dividing into two such paths, we eliminate the recalculation of the gradient; therefore, the speed of training is increased. Moreover, splitting into two paths, with each path being a part taken from the previous feature map, the number of parameters is also significantly reduced, thereby speeding up the whole inference process.Neck: The neck is responsible for mixing and matching the feature maps learned through the feature extraction (backbone) and identification process (YOLOv4, called Dense prediction). YOLOv4 allows customization using the following Neck structures: FPN (Feature Pyramid Networks) [40], PAN (Path Aggregation Networks) [41], NAS-FPN (Neural Architecture Search–Feature Pyramid Networks) [42], Bi-FPN (Bidirectional feature pyramid network) [43], ASFF (Adaptively Spatial Feature Fusion) [44], SFAM (Scale-wise Feature Aggregation Module) [45], SSP (spatial pyramid pooling layer) [46]. In the latter, SSP is a CNN network but is slightly changed; it is no longer about dividing feature maps into bins and then concatenating these bins together to obtain a fixed-dimensional vector.SPP, whose input is a feature map, outputs C×H×W from the backbone before being fed to the fully-connected layer to perform detection; YOLO applies the spatial pyramid pooling layer to the feature map three times—that is, using the SPP block, as illustrated in Figure 9.Yolo-SPP applies a maximum pool with kernels of different sizes. The size of the input feature map is preserved, and the feature maps obtained from applying the max pool (with different kernel sizes) will be concatenated. The architecture of YOLO-SPP is shown in Figure 10. Yolov4 also re-applies this technique.Dense prediction: using one-stage detectors; Sparse Prediction: using two-stage detectors such as R-CNN.

YOLOv5 (v6.0/6.1) [28] has almost the same architecture as YOLOv4 [27] and includes the following components: CSP-Darknet53 as a backbone, SPP and PANet in the model neck, and the head used in YOLOv4. In YOLOv5 (v6.0/6.1), SPPF has been used, which is just another variant of the SPP block, to improve the speed of the network and apply the CSPNet strategy on the PANet model.

Backbone: YOLOv5 improves YOLOv4’s CSPResBlock into a new module, with one less Convolution layer than YOLOv4, called the C3 module. Activation function: YOLOv4 uses the Mish or LeakyReLU for the lightweight version, while in YOLOv5, the activation function used is the SiLU.Neck: YOLOv5 adopts a module similar to SPP but twice as fast and calls it SPP-Fast (SPPF). Instead of using parallel max-pooling as in SPP, YOLOv5 SPPF uses sequential max-pooling, as illustrated in Figure 11. The kernel size in SPPF’s max-pooling is 5 instead of 5, 9, 13, as in YOLOv4’s SPP. Therefore, Neck in YOLOv5 uses SPPF + PAN.Other changes in YOLOv5 include the following:
-Data Augmentation techniques applied in YOLOv5 include Mosaic Augmentation, Copy–paste Augmentation, and MixUp Augmentation.-Loss function: YOLOv5 uses three outputs from PAN Neck to detect objects at three different scales. However, the effect of objects at each scale on Objectness Loss is different; so, the formula for Objectness Loss is changed to Equation (Equation 9).
(9)$${\mathrm{LF}}_{object}=4.0*{\mathrm{LF}}_{object}^{small}+1.0*{\mathrm{LF}}_{object}^{medium}+0.4*{\mathrm{LF}}_{object}^{large}$$-Anchor Box (AB): AB in YOLOv5 received two major changes. The first is to use auto anchor, a technique that applies Genetic Algorithms (GA) to the AB after the k-means step so that the AB works better with custom datasets, and not only works well on the MS COCO dataset. The second is to offset the center of the object to select multiple ABs for an object.

**Figure 11 sensors-23-03255-f011:**
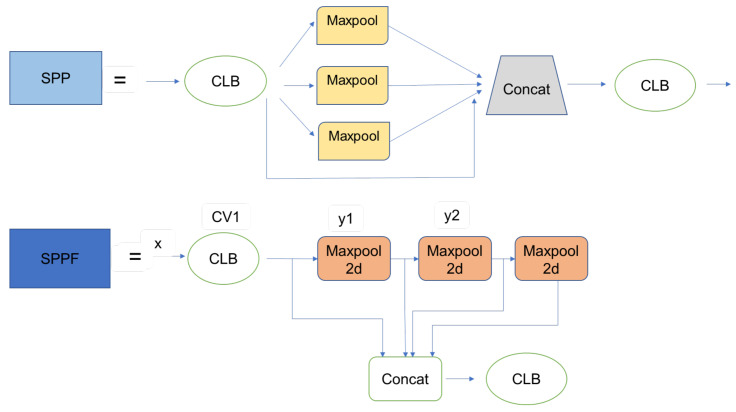
Comparison of the architecture of SPP and SPPF [47].

YOLOv7 [9], just like other versions of YOLO, consists of three parts in its architecture, as shown below:Backbone: ELAN, E-ELAN;Neck: CSP-SPP and (ELAN, E-ELAN)-PAN;Head: YOLOR [11] and Auxiliary head.

YOLOv7 includes some major improvements. First, the Efficient Layer Aggregation Networks (ELAN) is proposed to expand to Extended Efficient Layer Aggregation Networks (E-ELAN), where the strategy that learns at more depth with the shortest and longest derivatives along the slope will have a higher probability of convergence. This means not changing the gradient transmission path of the original architecture but increasing the group of convolutional layers of the added features and combining the features of different groups by mixing and merging the cordiality manner, as presented in Figure 12. This way of working can improve the learning efficiency of learned solid maps and improve the use of parameters and calculations. This process increases the accuracy of the learned model without increasing complexity and computational resources.

Second is the proposed Model Scaling for Concatenation-based Models (MSCM). The main idea of MSCM is based on scaled-YOLOv4 [13] to adjust the number of stages. When increasing the depth of a translation layer, which is immediately after, a concatenation-based computational block will increase, as illustrated in Figure 13a,b. It means the input width of the subsequent transmission layer increases. Therefore, the model scaling on concatenation-based models is proposed. This process only requires the depth in a computational block to be scaled, and the remaining transmission layer is performed with corresponding width scaling, as illustrated in Figure 13c.

The third is to reduce the number of parameters and computation for object detection. YOLOv7 is re-parameterized to combine with a different network. This work can reduce about 40% of the parameters and 50% computation of the object detector, and the detection will be faster and more accurate.

The fourth is a new label assignment method—as illustrated in Figure 14c,d—that guides both the auxiliary head and lead head by the lead head prediction. This method uses lead head prediction as a guidance to generate coarse-to-fine hierarchical labels, as illustrated in Figure 14e.

### 3.2. Comparative Study for Hand Detection and Classification

In this paper, we perform a comparative study on YOLO-family networks for hand detection and classification of the EV datasets. The taxonomy of the comparative study is illustrated in Figure 15. In this study, the methods are the YOLO-family networks whose development and improvements are presented in Section 3.1. Two models developed from the YOLO-family networks are hand detection and classification. The hand detection model is the main model tested on all YOLO versions; the hand classification model is only tested from YOLOv3 and later. The datasets used to evaluate the two models are FPHAB, HOI4D, and RehabHand, as presented in Section 4.1. The FPHAB database performs hand detection model evaluation and classifies action hands, background, and other objects. The HOI4D and RehabHand datasets perform the hand detection model assessment and classify left hand, right hand, background, and other objects. In Figure 15, we also present a comparative study of measures and outputs, described in Section 4.2 and Section 4.3.

## 4. Experimental Results

### 4.1. Datasets

The FPHAB dataset [4] is the First Person Hand Action Benchmark (FPHAB). This dataset is captured from an Intel RealSense SR300 RGB-D camera attached to the shoulder of a person. The resolutions of the color and depth images are 1920×1080 pixels and 640×480 pixels, respectively. The hand pose is captured using six magnetic sensors; it provides 3D hand pose annotation and intrinsic parameters for converting 2D hand pose annotation. There are several subjects (6 in total) performing multiple activities from 3 to 9 times with 45 hand actions. The number of joints in each 3D hand pose is 21. From attaching the device to mark 3D hand annotation data, the hand data have different characteristics compared to normal hands when obtained from EV. In this paper, we used configurations for training and testing, presented as follows: The configuration (Conf.#123) used the first sequence in each subject from Subj.#1 to Subj.#6 for testing (27,097 samples), the second sequence in each subject for validation (25,475 samples), and the remaining sequence for training (52,887 samples) (the ratio is approximately 1:2.5 for testing and training the model).

The HOI4D dataset [8] is collected and synchronized based on the Kinect v2 RGBD sensor and the Intel RealSense D455 RGB-D sensor. This is a large-scale 4D EV dataset with rich annotation for category-level human–object interaction. HOI4D includes 2.4M RGB-D frames of EV with over 4000 sequences. It is collected from 9 participants interacting with 800 different object instances from 16 categories over 610 different indoor rooms. This dataset provides ground-truth data of the following types: Frame-wise annotations for panoptic segmentation, motion segmentation, 3D hand pose, category-level object pose, and hand action, together with reconstructed object meshes and scene point clouds. The annotation data components are illustrated in Figure 16. To obtain the ground-truth data, which is the bounding boxes of the hand on the image for evaluation, we rely on the hand keypoints named “kps2D” in the 3D hand pose annotation. We take the bounding box of 21 hand keypoint annotations. This process is demonstrated in the source code of the “*get_2D_boundingbox_hand_anntation.py*” file found at the following link (https://drive.google.com/drive/folders/1yzhg5NsalPkOHI6CMkAE07yv5rY63tI7?usp=sharing, accessed on 30 January 2023).

The RehabHand dataset [17] was collected from rehabilitation exercises of patients at Hanoi Medical University Hospital, Vietnam. This dataset consists of frames from the first-person video captured by cameras worn by the patient on the forehead and the chest. The videos are recorded with a 1080p resolution at 30 frames per second. The data were collected using the GoPro Hero4 camera in San Mateo, California, USA. The camera recorded the exercise of 15 patients performing four upper extremity rehabilitation exercises. Each patient performed each exercise five times. The content of exercises related to grasping objects in different positions is presented as follows: exercise 1—practice with the ball, exercise 2—practice with a water bottle, exercise 3—practice with a wooden cube, exercise 4—practice with round cylinders. The collected data include 10 video files in MPEG-4 format with a total duration of 4 h and a total capacity of 53 GB recorded. The data are divided into three subsets for the training set (2220 images), validation set (740 images), and testing set (740 images) with a ratio of 6:2:2, respectively. Figure 17 illustrates the image data of the RehabHand dataset [17].

In this paper, we used a server with a NVIDIA GeForce RTX 2080 Ti 12 GB GPU for fine-tuning, training, and testing. The programs were written in the Python language (≥3.7 version) with the support of CUDA 11.2/cuDNN 8.1.0 libraries. In addition, there are a number of other libraries such as OpenCV, Numpy, Scipy, Pillow, Cython, Matplotlib, Scikit-image, Tensorflow ≥ 1.3.0, etc.

### 4.2. Evaluation Metrics

Similar to the evaluation of object detection and classification on images, we perform the calculation of the IOU (Intersection over Union) value according to Equation (Equation 10).
(10)$$IOU=\frac{B_{g}\cap B_{p}}{B_{g}\cup B_{p}}$$
where $B_{g}$ is the ground truth bounding box of hand action and $B_{p}$ is the predicted bounding box of hand.

To determine whether the bounding box is a true finding, we use a threshold $Thesh_{IOU}$ for the evaluation. If IOU is greater than or equal to $Thesh_{IOU}$, it is a true detection; otherwise, it is false.

In this paper, we also distinguish between the hand action, left hand, right hand, and the background; so, we also use the formulas for precision (*P*), Recall (*R*), and F1-Score (*F*1) (Equation (Equation 11)) to evaluate the analysis results of hand action classification on the image.
(11)$$P=\frac{TP}{TP+FP};R=\frac{TP}{TP+FN};F1=\frac{2⋆(R⋆P)}{\left(R+P\right)}$$
where TP are True Positives, TN are True Negatives, FP are False Positives, and FN are False Negatives. In addition, we also evaluate mAP.5 (mean Average Precision), computed as Equation (Equation 12).
(12)$$mAP=\frac{\sum_{i=1}^{c}AP_{i}}{c}$$
where averaging the average precision (AP) for all classes involved in the trained model yields mAP.

We train YOLO-family networks with 50 epochs and batch size = 4 frames; the size of the image can be img_size=640×640 or img_size=1280×1280, conf_thres=0.001. The hyper-parameter in the feature-extraction phase that the YOLO-family networks uses is the adaptive moment estimation (ADAM) optimizer [48], the learning rate is 0.001, and momentum is 0.937, as illustrated in Figure 18. There are also some other parameters shown in Table 1.

In this paper, we re-trained the YOLO-family networks (YOLOv4-CSP [13], YOLOv4-CSP-X [13], YOLOv3 [26], YOLOv3-SPP [49], YOLOv4 [13], YOLOv5-r50-CSP [28], YOLOv5-X50-CSP [28], YOLOv7 [9], YOLOv7-X [9], YOLOv7-w6 [9]) on the training set of Conf.#123 of the FPHAB dataset, the training set of the HOI4D dataset, and the training set of the RehabHand dataset. After that, we evaluated it on the validation set and testing set of configuration Conf.#123 of the FPHAB dataset, testing set of the HOI4D dataset, and testing set of the RehabHand dataset. We use the $Thesh_{IOU}$ to evaluate as follows: 0.5,0.75,0.95.

### 4.3. Hand Detection and Classification Results

The result of hand action detection and classification on the Conf.#123 of the FPHAB dataset is shown in Table 2. In the FPHAB dataset is the process of detecting the hand action in the image. The action hand detection and classification results on the FPHAB dataset in Table 2 of YOLOv7 and its variants are all greater than 95%. This is a very good result for the following steps on hand activity estimation and recognition.

In Table 2, it can be seen that the hand action detection and classification results on the FPHAB dataset are very accurate; the results are greater than 95%, even if the $Thesh_{IOU}=0.95$, which is close to absolute accuracy. Table 2 also shows that *P* is usually greater than *R* in most cases. This is because in the image of the FPHAB dataset, there can be two hands and, as a result, there are many background areas that are mistakenly detected as the hand action, so FN increase. Therefore, *R* is smaller than *P* in many cases. The processing time of the hand action detection and classification process is shown in Table 1; it is also very fast to ensure the pre-processing step without much impact on the processing time of the construction applications.

Figure 19 shows the results on precision, recall, F1-score, and confusion matrix on the hand action detection on the testing set of FPHAB dataset when $Thesh_{IOU}=0.5$.

Figure 20 shows the confusion matrix on classifying hand action on the testing set of the FPHAB dataset when $Thesh_{IOU}=0.5$.

Figure 21 illustrates some results of hand action detection and classification on the testing set of Conf.#123 of the FPHAB dataset when $Thesh_{IOU}=0.5$.

The results of hand detection and classification on the HOI4D dataset [8] are shown in Table 3. Table 3 shows the results of YOLOv7-w6 with the best results (R=89.85%;P=90.55%;mAP@.5=88.9%) when $Thesh_{IOU}=0.95$. This is a large dataset with many hand actions, for which the YOLO-family networks still obtain high results even when $Thesh_{IOU}=0.95$. In this dataset, the YOLO-family networks perform two tasks: detecting and classifying left and right hands. At the same time, the average result (all) of the left and right hands are also computed.

Figure 22 illustrates the results of hand classification on the HOI4D dataset based on YOLOv7. In Figure 22, there are many cases where the subject has the same color as the skin of the hand. However, YOLOv7 still detects and correctly classifies the hand.

The results of hand detection and classification on the RehabHand dataset [17] are shown in Table 4. The results in Table 4 show that YOLOv7 has the best results in detecting and classifying with the left hand (P=100%;R=92.1%;mAP@.5=14% with $Thesh_{IOU}=0.95$). YOLOv7-X has the best results in detecting and classifying with the right hand (P=87.7%;R=92.5%;mAP@.5=96.7% with $Thesh_{IOU}=0.95$), and the average result is also computed. It can be seen that the left hand detection results in some networks are very low because the left hand is as false negative as the right hand, as shown in Table 4.

Figure 23 illustrates the left hand being negatively classified as the right hand of the RehabHand dataset [17].

The results in Table 4 also show that the RehabHand dataset [17] is very challenging for hand detection and classification. This is a good dataset for evaluating hand detection models, hand pose estimation, and hand activity recognition.

## 5. Conclusions and Future Works

Building an application to evaluate the rehabilitation process of the hand using the technology of computer vision and deep learning is a new research area in the medical field. The first step is hand detection, which is a very important pre-processing step. In this paper, we systematize a series of versions of YOLO. We pre-trained hand detection and classification with versions of YOLO on the EV datasets FPHAB, HOI4D, and RehabHand. The results show the performance of the YOLO versions for hand detection and classification. All new versions of YOLO give better results than old versions. The results of YOLOv7 of hand detection and classification on the FPHAB dataset are the best (P=96.9% with $Thesh_{IOU}=0.5$, P=96.9% with $Thesh_{IOU}=0.75$, P=93.9% with $Thesh_{IOU}=0.95$). We apply this model to limit the hand data area, hand pose estimation, and hand activities recognition for evaluation hand function rehabilitation. YOLOv7 and its variations’ (YOLOv7-X, YOLOv7-w6) results on the HOI4D and RehabHand datasets are lower (Table 3 and Table 4) and unequal (Table 4). We perform pre-training with more epochs and calibrate the model’s parameter set to obtain a better model. Further, we compare YOLOv7 with CNN networks such as SSD, Faster R-CNN, and SOTA (State-Of-The-Art) on three datasets: FPHAB, HOI4D, and RehabHand. In the future, we will perform hand detection and tracking, hand pose estimation, and hand activity recognition for assessing the ability of the hand from faculty rehabilitation exercises of patients at Hanoi Medical University Hospital, Huong Sen Rehabilitation Hospital in Tuyen Quang Province in Vietnam [50], as illustrated in Figure 24.

## Figures and Tables

**Figure 1 sensors-23-03255-f001:**
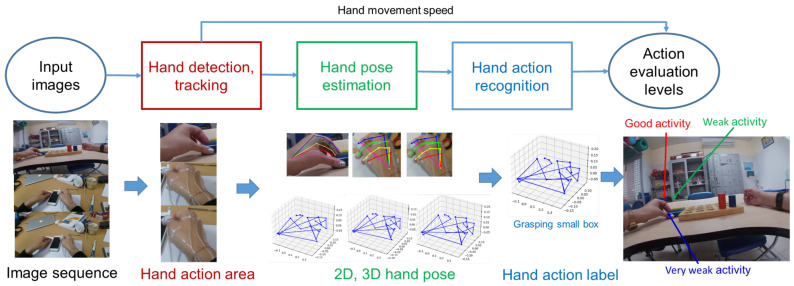
Framework for building an image-based rehabilitation evaluation system of the EV. Hand detection and classification is an important pre-processing step to limit the hand data area for hand pose estimation and activity recognition to assess hand activity levels.

**Figure 2 sensors-23-03255-f002:**
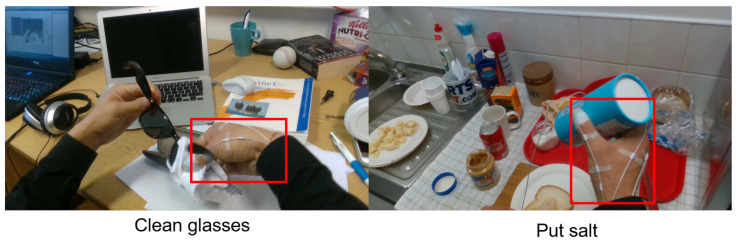
Illustration of obscured fingers in the FPHAB dataset [4].

**Figure 3 sensors-23-03255-f003:**
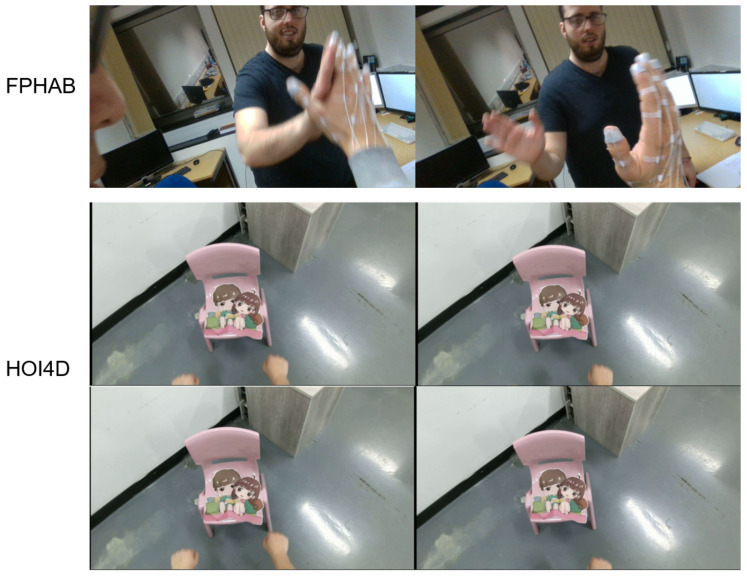
Illustrating some cases where the hand cannot be detected in the image when using the GM on the FPHAB and HOI4D datasets.

**Figure 4 sensors-23-03255-f004:**
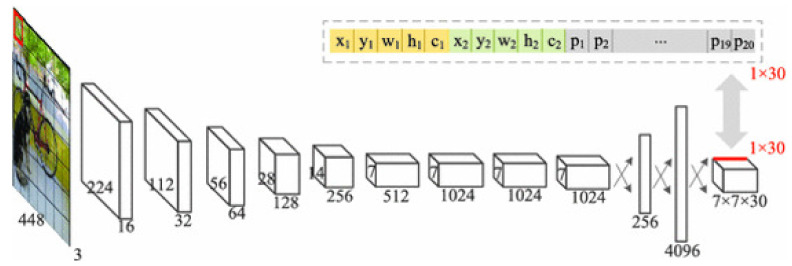
YOLOv1 architecture for object detection [24].

**Figure 5 sensors-23-03255-f005:**
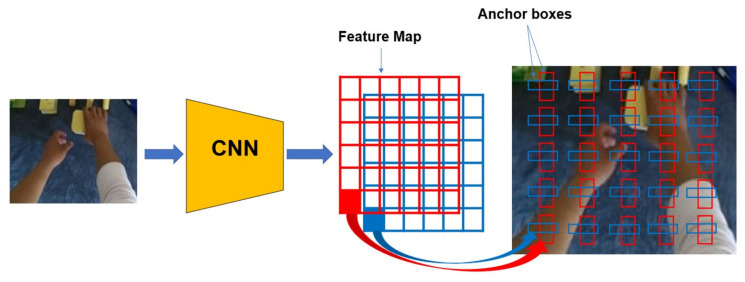
Anchor-based object detector.

**Figure 6 sensors-23-03255-f006:**
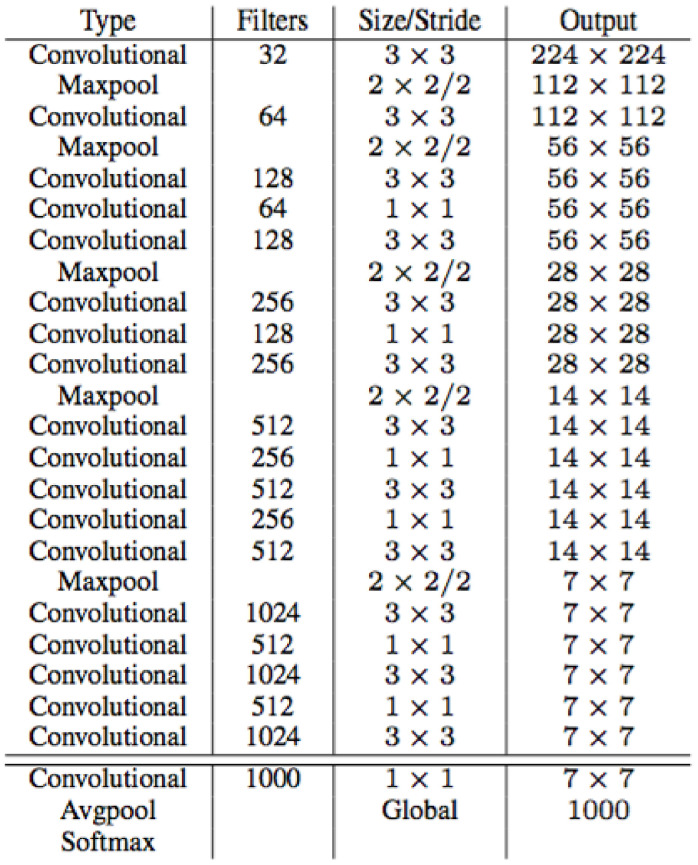
Darknet19 architecture.

**Figure 7 sensors-23-03255-f007:**
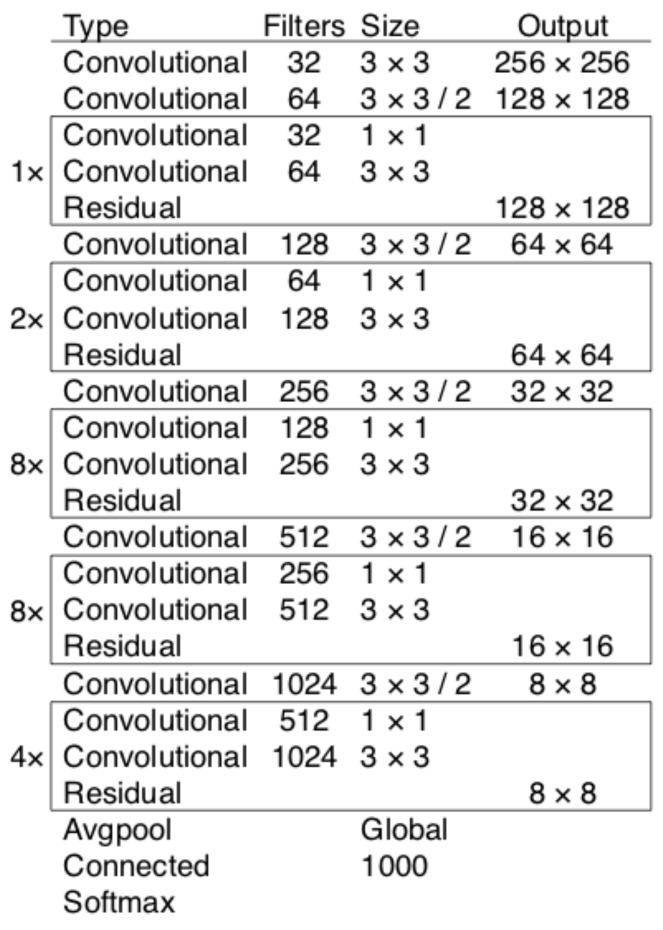
Darknet53 architecture.

**Figure 8 sensors-23-03255-f008:**
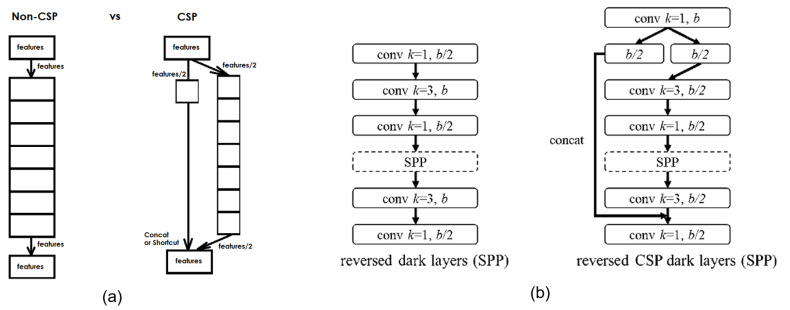
The architecture of CSPBlock in CSPDarknet53 [27]. (**a**) The simple CSP connection. (**b**) A CSP connection in YOLOv4-CSP/P5/P6/P7 [13].

**Figure 9 sensors-23-03255-f009:**
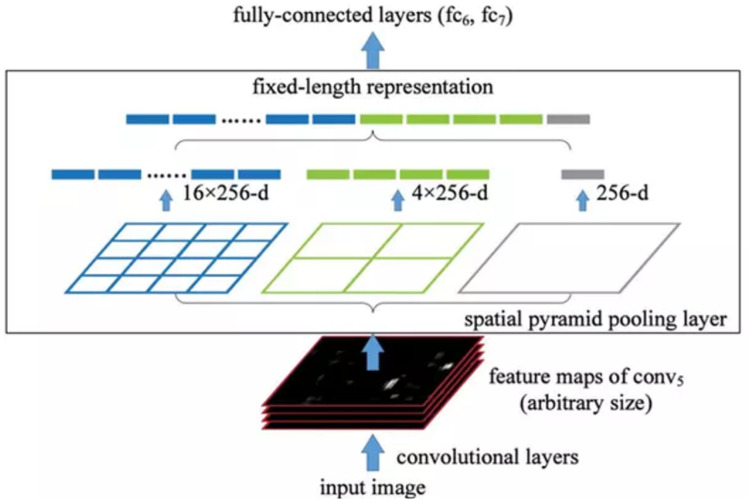
Architecture of SPP [13].

**Figure 10 sensors-23-03255-f010:**
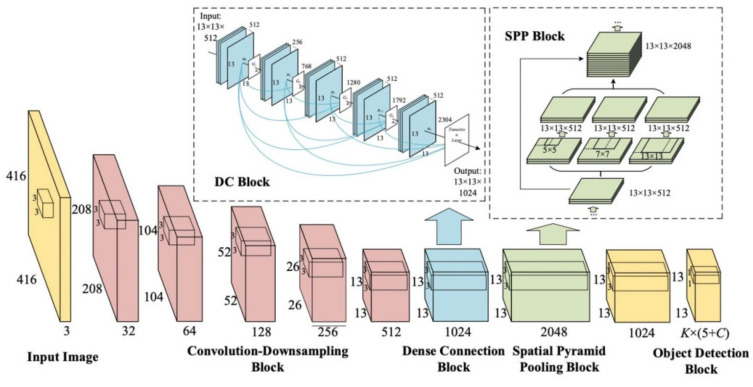
The architecture of YOLO-SPP bypasses the DC Block part [46].

**Figure 12 sensors-23-03255-f012:**
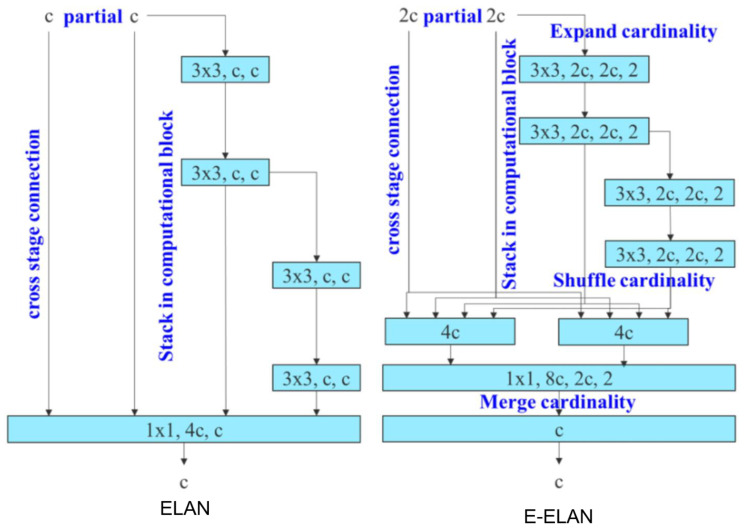
The architecture of ELAN and E-ELAN for efficient learning and faster convergence [9].

**Figure 13 sensors-23-03255-f013:**
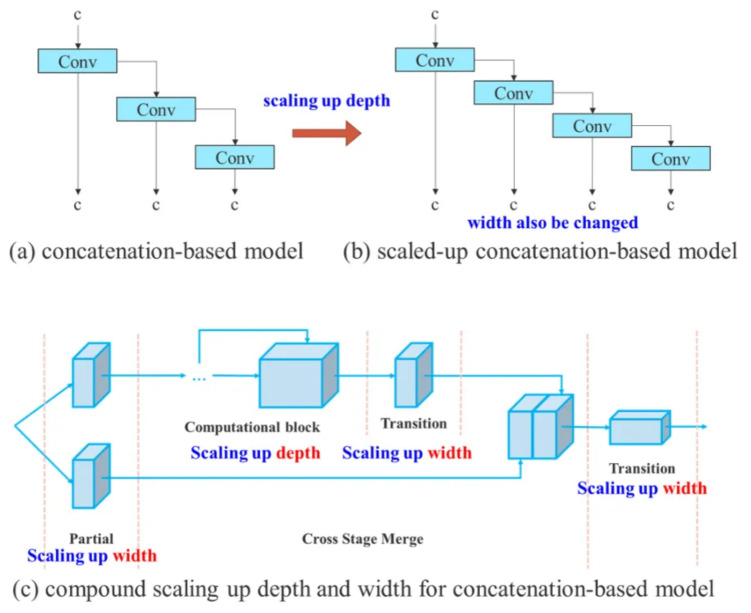
Illustrating of model scaling for concatenation-based models [9].

**Figure 14 sensors-23-03255-f014:**
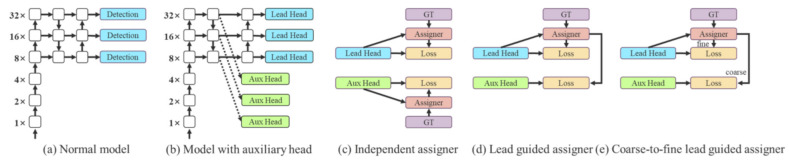
Illustration of coarse for auxiliary and fine for lead head label assigner [9].

**Figure 15 sensors-23-03255-f015:**
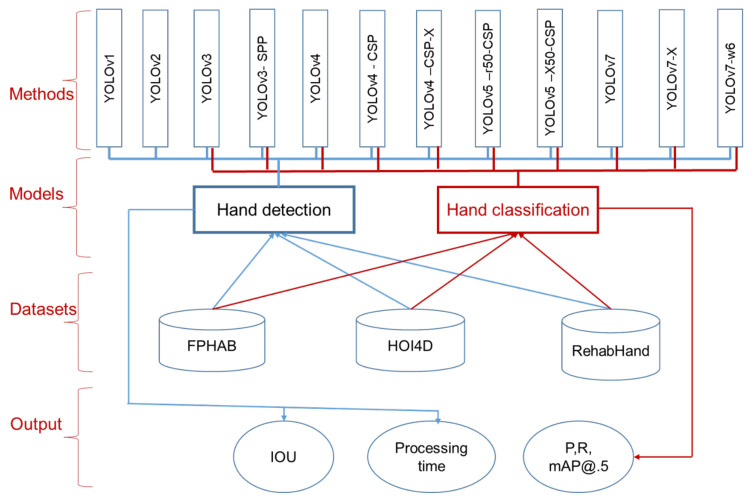
The taxonomy of the comparative study for hand detection and classification is based on the YOLO-family networks.

**Figure 16 sensors-23-03255-f016:**
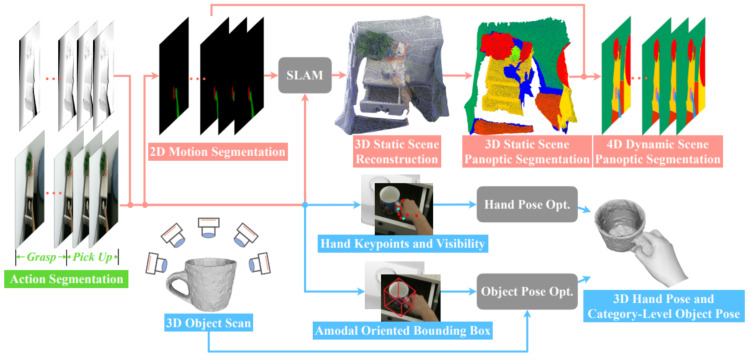
Describing the types of annotation data of the HOI4D dataset [8].

**Figure 17 sensors-23-03255-f017:**
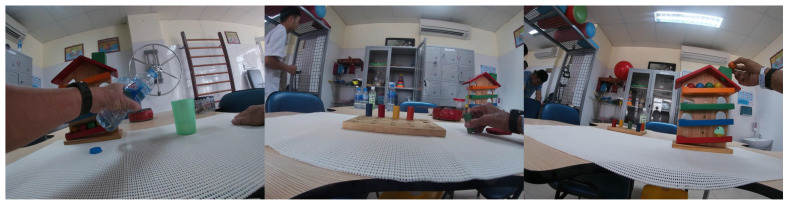
Illustrating the RGB image data obtained from the EV of the RehabHand dataset [17].

**Figure 18 sensors-23-03255-f018:**
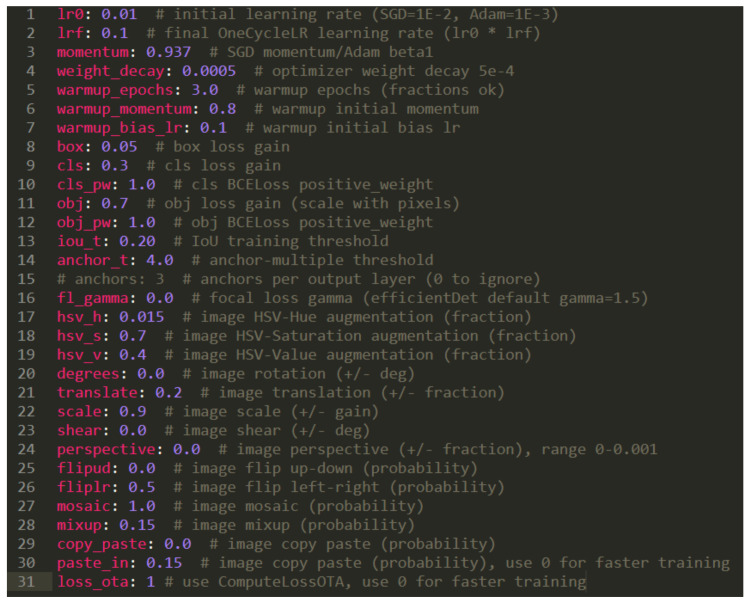
Illustrating the hyper-parameters of YOLO-family networks.

**Figure 19 sensors-23-03255-f019:**
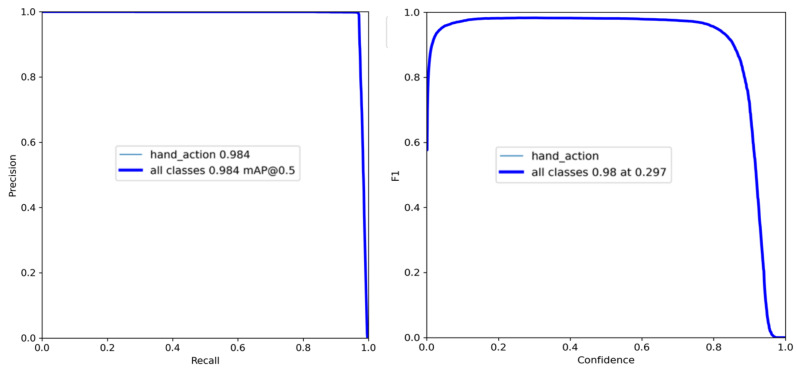
The distribution of precision, recall, and F1-score of hand action detection on the test set of Conf.#123 of the FPHAB dataset when $Thesh_{IOU}=0.5$.

**Figure 20 sensors-23-03255-f020:**
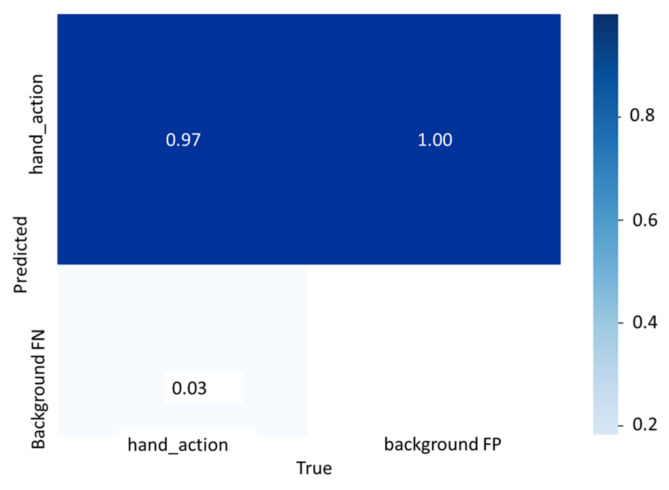
The confusion matrix of hand action classification on the testing set of Conf.#123 of the FPHAB dataset when $Thesh_{IOU}=0.5$.

**Figure 21 sensors-23-03255-f021:**
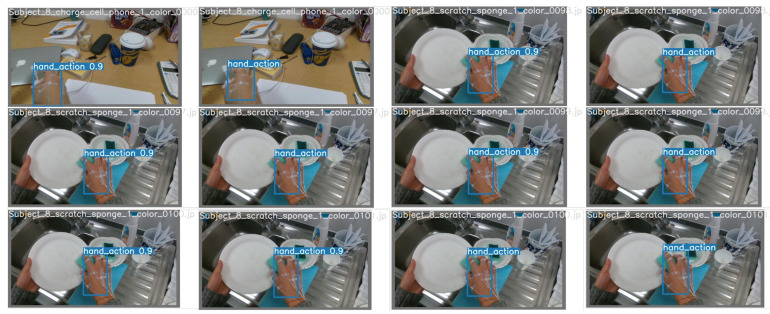
Illustrating some results of hand action detection and classification on the testing set of Conf.#123 of the FPHAB dataset when $Thesh_{IOU}=0.5$.

**Figure 22 sensors-23-03255-f022:**
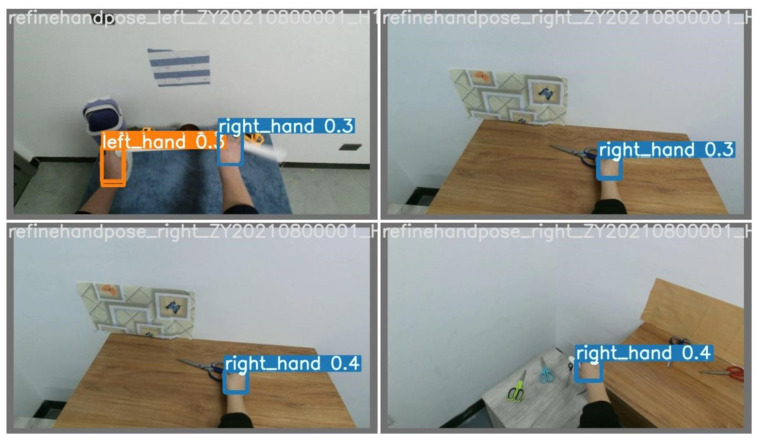
Illustration of hand classification results on the HOI4D dataset.

**Figure 23 sensors-23-03255-f023:**
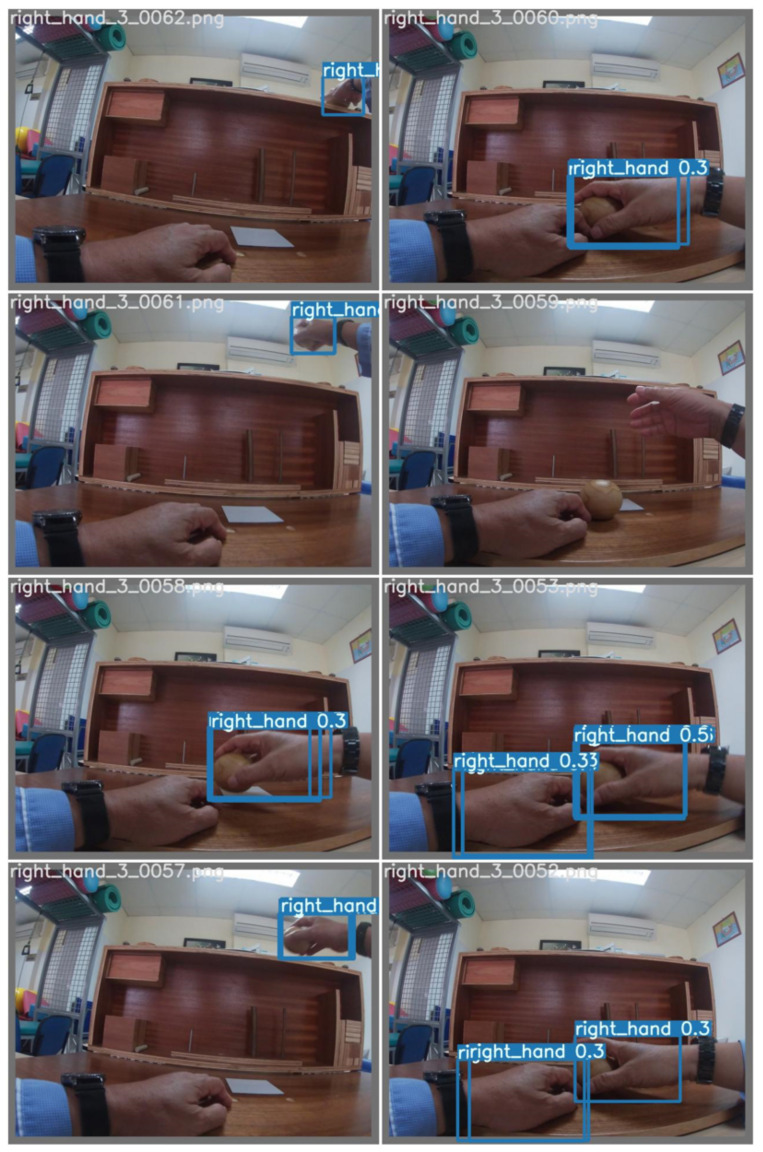
Illustrating the left hand being negatively classified as the right hand of the RehabHand dataset [17].

**Figure 24 sensors-23-03255-f024:**
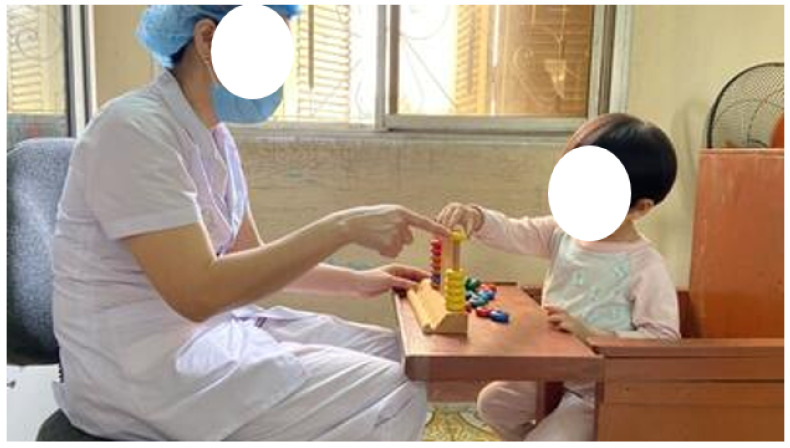
Illustrating the process of the hand rehabilitation exercise [50].

**Table 1 sensors-23-03255-t001:** The list of parameters of YOLOv7 and its variants [9], resulting in the processing time of the networks when evaluated on the testing set of the FPHAB dataset.

Methods	Image Size (pixel)	Number of Layers	Number of GFLOPS	Parameters	Number of Epochs	Processing Time for Testing (fps)
YOLOv4-CSP [13]	640 × 640	401	118.9	52,469,023	50	76.9
YOLOv4-CSP-X [13]	640 × 640	493	224.8	96,370,166	50	44
YOLOv3 [26]	640 × 640	261	154.5	61,497,430	50	153
YOLOv3-SPP [49]	640 × 640	269	155.4	62,546,518	50	142
YOLOv4 [13]	640 × 640	401	118.9	52,463,638	50	151
YOLOv5-r50-CSP [28]	640 × 640	314	103.2	36,481,772	50	133
YOLOv5-X50-CSP [28]	640 × 640	560	64.4	33,878,846	50	45
YOLOv7 [9]	640 × 640	314	103.2	36,481,772	50	133
YOLOv7-X [9]	640 × 640	362	188.0	70,782,444	50	98
YOLOv7-w6 [9]	1280 × 1280	370	101.8	80,909,336	50	60

**Table 2 sensors-23-03255-t002:** The results of hand detection and classification on the FPHAB dataset when performed on YOLOv7 and YOLO-family networks.

IOU Threshold(*Thesh_IOU_*)/ Precision(P)/ Recall(R)/ Models	0.5			0.75			0.95		
	R (%)	P (%)	mAP@.5 (%)	R (%)	P (%)	mAP@.5 (%)	R (%)	P (%)	mAP@.5 (%)
YOLOv4-CSP [13]	99.8	96.7	98.5	99.7	96.7	98.6	91.4	92.3	94.9
YOLOv4-CSP-X [13]	99.3	96.5	97.2	99.5	96.1	97.2	87.4	90.2	91.8
YOLOv3 [26]	99.9	96.6	98.5	99.9	96.6	98.8	96.4	95.1	97.5
YOLOv3-SPP [49]	99.5	97.6	98.8	99.6	97.3	98.9	95.3	92.2	97
YOLOv4 [13]	99.6	96.4	97.2	99.6	96.2	97.5	84.7	92	92.3
YOLOv5-r50-CSP [28]	99.3	95.2	98.1	99.5	96.5	98.1	92.5	91.4	93.5
YOLOv5-X50-CSP [28]	99.2	94.8	97.6	99.4	96.4	98.3	96.4	92.6	94.1
YOLOv7 [9]	99.7	96.9	98.7	99.4	96.9	99.4	97	93.9	98.2
YOLOv7-X [9]	99.2	98.7	99.1	99.2	98.2	99.5	97.5	94.5	97
YOLOv7-w6 [9]	99.7	96.9	99.8	99.7	96.9	99.7	93.9	98	98.3

**Table 3 sensors-23-03255-t003:** The results of hand detection and classification on the HOI4D dataset [8].

IOU Threshold/ (*Thesh_IOU_*) Precision(P)/ Recall(R)/ Models	Hand	0.5			0.75			0.95		
		R (%)	P (%)	mAP@.5 (%)	R (%)	P (%)	mAP@.5 (%)	R (%)	P (%)	mAP@.5 (%)
YOLOv4-CSP [13]	Right hand	90.7	96.2	95.2	96.4	90.6	95.1	88.9	90.1	91.7
	Left hand	82.5	86	85.5	82.2	85	84.7	76.8	65.4	74.4
	All	88.4	89.4	90.3	89.3	87.8	89.9	82.8	77.7	83.1
YOLOv4-CSP-X [13]	Right hand	96.2	90.7	95	96.2	90.6	94.9	84.2	90.2	90.9
	Left hand	81	86	84.8	82	83.3	84.1	76	62.9	71.9
	All	88.6	88.3	89.9	89.3	87	89.5	80.1	76.6	81.4
YOLOv3 [26]	Right hand	89.9	90.8	94	89.9	90.8	94.2	77.4	89.3	90.7
	Left hand	81.5	81.8	82.4	80.4	81.9	81.6	75.1	61.8	71.2
	All	85.7	86.3	88.2	85.1	86.3	87.9	76.2	75.6	80.9
YOLOv3-SPP [49]	Right hand	88.3	90.8	94.1	89.2	90.7	94.2	69.2	88.8	86
	Left hand	81.6	81.2	82.1	81.4	79.9	81.1	70.3	58.7	65.4
	All	84.9	86	88.1	85.3	85.3	87.6	69.7	73.8	75.7
YOLOv4 [13]	Right hand	89.7	93.4	95.9	90.5	94.6	95.9	71.5	88.8	89.4
	Left hand	82.8	83.5	84.3	84.2	82.4	86.3	72.4	78.3	75.2
	All	86.3	88.5	88.1	87.4	88.5	91.1	72	83.6	82.3
YOLOv5-r50-CSP [28]	Right hand	84.3	87.5	90.9	84.4	87.4	90.9	63	81.2	78.4
	Left hand	79.4	77.6	78.7	78.8	76.9	78.4	61.5	53.6	57.5
	All	81.9	82.5	84.8	81.6	82.1	84.6	62.2	7.4	68
YOLOv5-X50-CSP [28]	Right hand	94.1	90.2	92.7	90.4	89.6	90.8	78.2	88.2	84.4
	Left hand	79.4	77.6	78.7	78.8	76.9	78.4	61.5	73.6	77.5
	All	86.75	83.9	85.7	84.6	83.25	84.6	69.85	80.9	80.95
YOLOv7 [9]	Right hand	87	90.7	93.3	81	90.7	93.4	69.6	89.3	86.4
	Left hand	81.4	78.9	80.7	81.3	78.8	80.8	61.7	56.1	60.8
	All	84.2	84.8	87	84.2	84.8	87.1	65.7	72.7	73.6
YOLOv7-X [9]	Right hand	91.1	90.6	94.1	91.6	90.6	94.2	74.1	89.6	88.4
	Left hand	80.7	81.1	81.2	80.2	80.2	80.6	65.4	59.1	64
	All	85.9	85.9	87.7	85.9	85.4	87.4	69.8	74.3	76.2
YOLOv7-w6 [9]	Right hand	99.3	97.7	97	97.4	94.7	98.7	92.8	94.7	93
	Left hand	86.7	92.3	95.1	85.5	89.3	88.8	86.9	86.4	84.8
	All	93	95	96.05	91.45	92	93.75	89.85	90.55	88.9

**Table 4 sensors-23-03255-t004:** The results of hand detection and classification on the RehabHand dataset [17].

IOU Threshold/ (*Thesh_IOU_*) Precision(P)/ Recall(R)/ Models	Hand	0.5			0.75			0.95		
		R (%)	P (%)	mAP@.5 (%)	R (%)	P (%)	mAP@.5 (%)	R (%)	P (%)	mAP@.5 (%)
YOLOv4-CSP [13]	Right hand	33.8	55.8	29.6	32.9	54.3	28.4	28.2	36.9	20.8
	Left hand	100	98.9	93.7	100	97.4	92.1	100	92.4	5.1
	All	66.9	77.35	63.45	66.45	75.85	60.25	64.1	66.2	12.95
YOLOv4-CSP-X [13]	Right hand	34.2	56	30.5	33.7	56	30.6	26.6	37.1	21.1
	Left hand	100	97.2	95.4	100	95.7	8.12	100	92.6	5.23
	All	67.1	76.6	62.95	66.8	75.85	19.3	63.3	64.85	13.2
YOLOv3 [26]	Right hand	2.41	94.1	18.9	0.0861	98.8	19.3	0.045	99	11.2
	Left hand	20.8	65.9	17.3	20.5	65.9	17.2	19.4	24.6	12.8
	All	11.6	80	18.1	10.7	82.4	18.3	9.91	61.8	12
YOLOv3-SPP [49]	Right hand	17.5	51.6	17.4	17.6	47	17.5	15.2	45	12.6
	Left hand	100	98.5	90.7	100	96.8	85.2	100	91.7	89.7
	All	58.8	75.5	54.05	58.8	97.9	53.6	58.8	68.35	51.15
YOLOv4 [13]	Right hand	2.26	91.8	21.3	0.91	97.7	21.2	16.4	37	14.5
	Left hand	23.9	55.7	21.9	22.2	55.7	20.8	100	25.8	10.5
	All	13.1	73.8	21.6	11.6	76.7	21	58.2	31.4	12.5
YOLOv5-r50-CSP [28]	Right hand	25.9	68.9	24.1	25.7	67.4	24	0.44	99.1	14.7
	Left hand	100	96.1	92.8	100	94.3	14.6	22.9	40.5	14.1
	All	63	82.5	57.4	62.9	80.85	19.3	11.7	69.8	14.4
YOLOv5-X50-CSP [28]	Right hand	1.69	95.8	19.1	0.62	99.5	17.1	0.33	99.3	6.84
	Left hand	24.2	53.5	23.6	24	53.6	53.6	27.7	23.3	18.9
	All	13	74.7	21.4	12.3	76.6	20.3	14	61.3	12.9
YOLOv7 [9]	Right hand	81.2	96.3	95.2	78.7	91.7	98.8	75.3	96.8	96.7
	Left hand	100	99.2	97.3	100	97.8	96.7	100	92.1	92.4
	All	90.6	97.75	96.25	89.35	94.75	97.75	87.65	94.45	94.55
YOLOv7-X [9]	Right hand	91.3	95.3	93.6	91.1	92.7	93.6	87.7	92.5	96.7
	Left hand	100	96.4	96.2	100	95.5	7.82	100	90.8	90.5
	All	95.65	95.85	95.45	95.55	94.1	50.71	93.85	92.3	93.6
YOLOv7-w6 [9]	Right hand	1.77	96.8	19.9	0.498	99.7	19.5	0.0169	99.5	9.53
	Left hand	25.8	55.5	15.4	24.3	55.6	14.8	15.2	32.9	5.62
	All	13.8	76.1	17.6	12.4	77.6	17.2	7.66	66.2	7.57

## Data Availability

Not applicable.

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
