# Peer review of "YOLO Series for Human Hand Action Detection and Classification from Egocentric Videos"

_sensors, 2023, doi:10.3390/s23063255_

Round 1

Author Response

We send you responses based on your comments and revised versions of the manuscript in the attachment

Reviewer 2 Report

Authors have worked for human hand action detection and classification from Egocentric videos using YOLO object detection family.

Abstract is very poor. It has to be enhanced.

It seems that page no. 3 is from result section and kept in between contributions. It should be updated.

Figure resolution needs to be enhanced. Image resolution must be at least 300 dpi.

Authors have explained about existing network a lot about all YOLO versions. It should be explained the major work done by the authors considering the YOLO versions. It looks like novelty is missing. Explain your novel work in section 3.

Figure 6 have been taken from other than research paper. It should be avoided and should draw by your own with proper tools, not like hand drawing.

The size of the figure 17 can be kept small.

Hyperparamters values needs to be mentioned.

Authors have compared work with YOLO family. It should also compare with SOTA methods.

English editing is highly recommended.

Author Response

(The authors gave the same response as above.)

Reviewer 3 Report

The paper tackles the issue of hand detection and classification from egocentric videos. Generally, the idea and motivation are good, and the topic is relevant and important. 

Nevertheless, the authors present different YOLO versions and datasets as the main contributions with the result that the newest version delivers the best result. Unfortunately, this is not really relevant research - obviously YOLO developers improve their system from version to version, which also manifests in the main conclusion of the paper: "New versions of YOLO all give good results"...

Author Response

(The authors gave the same response as above.)

Round 2

Reviewer 1 Report

Recommend for publication.

Reviewer 2 Report

Authors have address the given comments.

Page no. 18,19,20, and 21 needs formatting. Lots of white space are seen. It should be avoided.